# A Meta-Analysis Study of SOD1-Mutant Mouse Models of ALS to Analyse the Determinants of Disease Onset and Progression

**DOI:** 10.3390/ijms24010216

**Published:** 2022-12-22

**Authors:** Maria Ciuro, Maria Sangiorgio, Giampiero Leanza, Rosario Gulino

**Affiliations:** 1Department of Biomedical and Biotechnological Sciences, Physiology Section, University of Catania, 95123 Catania, Italy; 2Department of Drug and Health Sciences, University of Catania, 95125 Catania, Italy; 3Molecular Preclinical and Translational Imaging Research Centre—IMPRonTE, University of Catania, 95125 Catania, Italy

**Keywords:** ALS epidemiology, ALS pathogenesis, amyotrophic lateral sclerosis, mathematical model, meta-analysis, mouse, superoxide dismutase 1, symptom onset

## Abstract

A complex interaction between genetic and external factors determines the development of amyotrophic lateral sclerosis (ALS). Epidemiological studies on large patient cohorts have suggested that ALS is a multi-step disease, as symptom onset occurs only after exposure to a sequence of risk factors. Although the exact nature of these determinants remains to be clarified, it seems clear that: (i) genetic mutations may be responsible for one or more of these steps; (ii) other risk factors are probably linked to environment and/or to lifestyle, and (iii) compensatory plastic changes taking place during the ALS etiopathogenesis probably affect the timing of onset and progression of disease. Current knowledge on ALS mechanisms and therapeutic targets, derives mainly from studies involving superoxide dismutase 1 (SOD1) transgenic mice; therefore, it would be fundamental to verify whether a multi-step disease concept can also be applied to these animal models. With this aim, a meta-analysis study has been performed using a collection of primary studies (n = 137), selected according to the following criteria: (1) the studies should employ SOD1 transgenic mice; (2) the studies should entail the presence of a disease-modifying experimental manipulation; (3) the studies should make use of Kaplan–Meier plots showing the distribution of symptom onset and lifespan. Then, using a subset of this study collection (n = 94), the effects of treatments on key molecular mechanisms, as well as on the onset and progression of disease have been analysed in a large population of mice. The results are consistent with a multi-step etiopathogenesis of disease in ALS mice (including two to six steps, depending on the particular SOD1 mutation), closely resembling that observed in patient cohorts, and revealed an interesting relationship between molecular mechanisms and disease manifestation. Thus, SOD1 mouse models may be considered of high predictive value to understand the determinants of disease onset and progression, as well as to identify targets for therapeutic interventions.

## 1. Introduction

Amyotrophic lateral sclerosis (ALS) is a progressive neurodegenerative disease characterised by selective loss of both upper and lower motor neurons (MNs), leading to respiratory failure within 2–5 years from symptom onset [1]. Despite significant advances in ALS research, the exact mechanism that leads to disease development has not been defined with certainty although it is generally accepted that ALS has a marked multifactorial nature where both genetic and non-genetic factors cooperate to cause the pathological microenvironment [2]. Thus, therapeutic options remain limited to symptomatic management, functional support, palliative care and two approved drugs, with marginal efficacy [3,4]. It is clear today that the pathological processes leading to ALS are intricate events requiring the use of a complex system approach instead of multiple reductionist models [5]. In particular, ALS epidemiology appears as a complex mixture of environmental risks and time-related factors acting on a pathogenic genetic background and followed by an automatic progression [6]. Approximately 10% of patients have a family history of disease, (fALS) and about 10–20% of autosomal dominant fALS cases are generated by mutations in the gene encoding for superoxide dismutase 1 (SOD1). Unfortunately, the mechanism by which mutant SOD1 induces MNs degeneration and then leads to the appearance of motor symptoms remains poorly understood [7,8,9].

Recent studies suggest that the development of ALS is the final product of a step-by-step pathological process, supporting the idea that a number of sequential steps need to arise before patients manifest the disease. Moreover, it seems likely that compensatory plastic phenomena could take place, at least in the early phase of etiopathogenesis. Thus, the appearance of motor deficits may probably occur only when this compensatory adaptation becomes unable to counteract the increasing neurodegeneration [10,11,12,13,14,15,16]. A mathematical model, previously established by Armitage and Doll, and used in cancer epidemiology [17], was adapted for the first time to ALS, demonstrating that ALS is a six-step disease; therefore, patients have to be sequentially exposed to six different risk factors prior to showing symptoms [18]. One or more of these steps are thought to include the genetic risk, because a number of at-risk mutations may play a major role in triggering the degenerative process. This hypothesis was tested by a recent population-based modelling study conducted on Italian and Irish ALS registers, concluding that patient cohorts with a known ALS-causing mutation require fewer steps than those without such variants [19].

A large amount of in vivo experiments aiming at understanding the pathophysiology of ALS and at validating therapeutic targets, has involved transgenic mice overexpressing a mutant SOD1 gene [20,21,22].

Supposing that, both in humans and animal models, ALS is the final result of several sequential cellular and molecular changes, the use of animal models could represent a valuable tool to follow and dissect the stepwise progression of the disease. Thus, the main purpose of the present work was to apply an already established mathematical approach to verify whether the etiopathogenesis of disease in ALS mice follows a multi-step model, as already found in large cohorts of human patients [18,19].

In order to verify this hypothesis, a large number of animals is required. Therefore, a meta-analysis study has been performed herein, selecting a sufficient number of primary studies including the most common transgenic mouse models overexpressing the following mutations of the SOD1 gene: G93A, G37R, G85R, and H46R. If the multi-step model is valid in animals, it will be possible to estimate the number of steps required to develop the disease and to compare this number with those reported in the populations of human patients, confirming the usefulness of ALS mouse models from a novel point of view.

Furthermore, as a contribution to the identification of at least some of the determinants of ALS etiopathogenesis, a deeper investigation of the data extracted from primary studies has been done. In particular, molecular mechanisms linked to ALS and their experimental modifications have been correlated with changes observed in the onset of disease and animal lifespan.

## 2. Results

### 2.1. Collection of Primary Studies

Primary studies that comply with the eligibility criteria described in methods were collected for subsequent data extraction and analysis (Table 1). In particular, PubMed searching yielded a total of 725 papers for all the mutations analysed (SOD1^G93A^, SOD1^G85R^, SOD1^G37R^, SOD1^H46R^). After exclusion of the articles that did not comply with the inclusion criteria, 137 papers were selected (please see the Reference List in Appendix A), and these studies involved a significantly large total number of mice to produce a reliable analysis of data. Precisely, for the analysis of incidence data, a total number of 3038 control SOD1 transgenic mice (without any other experimental manipulation) were found included in the selected primary studies, with an average number of 22 mice per paper.

Mice have been also subdivided by sex (674 female and 633 male mice), and separate analyses of onset have been performed, with no significant differences (Mann-Whitney U test: *p* > 0.05). Moreover, the majority of primary studies did not declare the sex of mice or have used a sex balanced experimental design (39% and 33%, respectively; total number of mice: 1731), whereas only 28% of primary studies showed separate analyses of male and/or female mice. Therefore, female and male mice were pooled together, and all data analyses can be considered as sex-balanced. Primary studies appeared also to be balanced in terms of genetic background of transgenic mice. In particular, 38% and 45% of studies involved SOD1 transgenic mice belonging to the B6SJL and C57BL/6 genetic background, respectively, whereas 17% of studies did not state this information. Moreover, it was found that: (i) SOD1^G85R^ transgenic mice were low-expressors [23,24]; (ii) SOD1^G37R^ mice showed 2–14-fold expression of the transgene; (iii) SOD1^H46R^ were high-expressors [25]; (iv) SOD1^G93A^ were high-expressors in 87% and low-expressors in 5% of primary studies, with the remaining 8% of studies not including this information [23,24].

For subsequent analysis of pathological mechanisms and effects of experimental manipulations, a subset of the described primary studies was used, involving only SOD1^G93A^ mice, and an additional number of 1647 treated SOD1^G93A^ mice (subjected to drug treatments or other experimental manipulations) were included, with a total number of 4685 SOD1 transgenic mice included in the study.

### 2.2. Analysis of Incidence Data Revealed a Multi-Step Disease Development

Data concerning the age at onset of SOD1 mice have been extracted from the primary studies and used to calculate an estimation of monthly incidence, as described in the methods. For this part of the study, only control animals (without any experimental manipulation) were used. Briefly, the number of animals starting to show motor symptoms in each month have been divided by the number of animals still being healthy in the same month. After that, the Armitage and Doll mathematical model has been applied, and the logarithmic values of the incidence at any time-point were correlated with the corresponding logarithmic values of the age at onset (Table 2, Figure 1). For each mutation, a different range was found for the mouse age at onset. A small number of animals showed a very earlier or later onset, but they were excluded from the study because of the very low number of cases, giving inconsistent incidence values. Similar considerations were raised during epidemiological studies involving human patients [18,19]. However, about 95% of the mouse sample has been included in the study: 2867 out of 3038 mice (please see Table 1 and Table 2). After a careful analysis, it was found that these outliers were almost uniformly distributed across the main genetic backgrounds (B6SJL and C57BL/6) and sexes, as well as concerning the level of transgene expression (low- or high-expressors).

A linear relationship has been found between log age and log incidence for all the SOD1 mutations analysed (Figure 1), which is consistent with a multi-step pathological process [18,19]. In particular, mice overexpressing the SOD1^G93A^ mutation showed an age at onset ranging from 2 to 6 months (average age at onset: 16.0 weeks), and a statistically significant linear correlation between log age and log incidence (R^2^ = 0.939; *p* = 0.007), with an estimated slope of 1.158 (Figure 1A, Table 2 and Table 3). The SOD1^G85R^ mouse population showed an age at onset ranging from 8 to 14 months (average age at onset: 46.4 weeks), with a significant correlation and linear regression (slope of 4.839) between log age and log incidence (R^2^ = 0.914; *p* = 0.001; Figure 1B, Table 2 and Table 3). The analysis of the SOD1^G37R^ sample confirmed a statistically significant linear relationship between log age and log incidence (R^2^ = 0.838; *p* = 0.000), with an estimated slope of 1.525 (Figure 1C, Table 3); the age at onset for these animals was found to range from 4 to 13 months, with an average value of 32.8 weeks (Table 2 and Table 3). The analysis of the population of mice overexpressing the SOD1^H46R^ mutation demonstrated, again, a linear relationship between log age and log incidence (R^2^ = 0.996; *p* = 0.001), with an estimated slope of 3.002 (Figure 1D, Table 3). These animals showed an age at onset ranging from 4 to 7 months with an average value of 21.6 weeks (Table 2 and Table 3).

These incidence data, as analysed by applying the Armitage and Doll model, suggest that the pathological mechanism preceding the onset of disease could be a multi-step process, and that the number of steps was different for the different SOD1 mutations considered. A comparison among the cited mutations is showed in Table 3, which reports the average values of age at onset and incidence, as well as the slope of the regression line and the corresponding estimated number of steps, as defined by the Armitage and Doll model. Interesting information may be obtained from these data, which is the apparent inverse relationship between the average log age at onset and the average log incidence measured in the mouse population (R^2^ = 0.905; *p* = 0.048; Table 3, Figure 2A).

It appears that the earlier the appearance of symptoms, the higher will be the incidence in a given mouse population. As a consequence, it seems that the higher the number of steps towards onset, the higher will be the age at onset of a given population of SOD1 transgenic mice, although this correlation was found to be not statistically significant (R^2^ = 0.691; *p* > 0.05; Figure 2B).

Finally, in order to validate the mathematical model, the predicted values of monthly incidence were calculated by using the equations associated with each linear model (Figure 1) and then correlated with the actual incidence values extracted from primary studies. Figure 2C shows the significant correlation between actual and predicted logarithmic incidence values (R^2^ = 0.933; *p* = 0.001), thus suggesting that the model is capable of predicting the incidence of motor symptoms in a specific SOD1 transgenic mouse population of a given age.

### 2.3. Extraction of Data concerning the Cellular and Molecular Mechanisms Associated with Disease Onset and Progression

The subset of primary studies involving SOD1^G93A^ mice (n = 109 studies; Table 1) were then considered for the extraction and analysis of data concerning the pathological mechanisms that are probably responsible for the determination of symptom onset, and also for the subsequent progression of disease. Some of these studies (n = 15) were excluded from this analysis due to one or more of the following conditions: (1) lack of information concerning the effects of experimental procedures on the age at onset or survival; (2) treatments starting after the onset of symptoms; (3) observational studies lacking a disease-modifying approach.

For each of the selected primary studies (n = 94, please find these papers in Appendix A), the following data were collected: (1) type of treatment or experimental manipulation (e.g., drug administration; gene silencing; gene overexpression); (2) the most frequently reported cellular or molecular mechanisms that were significantly modified (e.g., cell death, muscle denervation, muscle atrophy, reactive gliosis or other immune responses); (3) effects on the age at onset (significant anticipation or delay); (4) effects on the animals’ lifespan (significantly reduced or increased). Given the diversity of experimental approaches and mechanisms that were described in the primary studies, they have been categorized as described in Table 4. In particular, considering the scientific hypotheses stated in each primary study, treatments and other experimental manipulations were classified based on the stated mechanism of action. Therefore, experimental processes capable of inhibiting stress mechanisms (e.g., oxidative stress, glutamate excitotoxicity; apoptosis and other related pathological processes), proteinopathy (including misfolding, aggregation and toxicity of SOD1 protein, or other misfolding proteins known to be associated with ALS, as well as related processes including autophagy or endoplasmic reticulum stress), immune mechanisms (including inflammation, reactive astrogliosis and microgliosis, leukocyte activation and so on) or processes regulating cell fate were classified in the corresponding category. Additionally, all treatments capable of stimulating trophic mechanisms were included in a category named “Trophic Support”. Lists of primary articles using the experimental approaches falling in the described category can be found in Appendix A. Please note that, except for trophic support, all other treatment categories are intended as inhibitory actions; in fact, when a primary study reported a stimulatory treatment, both treatment and its effects on controlled mechanisms have been logically inverted, in order to make a consistent body of data that can be analysed together. Therefore, all data reported below, regarding the link between onset (or survival) and the underlying mechanisms, or the effects of treatments on these processes, are intended as the effects of inhibiting stress, proteinopathy, immune response, cell fate mechanisms, and stimulating trophic support.

Regardless of the specific target and mechanism of action, each treatment (drug treatment or other experimental protocol), may produce significant effects on multiple mechanisms (Table 4 and Appendix A). Among these, the most frequently reported cellular and molecular mechanisms that can be modified by treatments and conversely influence functional outcome (including onset and survival), can be included in the broad categories of Immune Response (see above) and Neuromuscular Degeneration (including motoneuron death, muscle denervation and muscle atrophy).

### 2.4. Effects of Experimental Procedures on Disease Onset and Survival

One of the aims of the present study was to evaluate if the experimental protocols (e.g., drug treatments, genetic manipulations or other approaches) that were supposed to act on a specific target, may be capable of modifying the age at onset or the animal’s lifespan. As described above, treatments have been classified in five different target categories and then correlated to the modifications of onset and survival observed in the same primary study (Figure 3).

This analysis has revealed that experimental procedures aiming at modifying specific mechanisms (as stated in the hypothesis of each primary study) have the following effects on the age at onset of SOD1^G93A^ animals and their lifespan: treatments aiming at reducing “Stress” processes produced a delayed disease onset (N = 23; Chi-square = 25.4; *p* < 0.001; Figure 3A) and an extended survival of mice (N = 23; Chi-square = 29.8; *p* < 0.001; Figure 3F).

On the other hand, inhibition of pathological mechanisms linked to “Proteinopathy” caused an extended survival (N = 16; Chi-square = 12.9; *p* < 0.005; Figure 3G) but they did not affect the disease onset (N = 16; Chi-square = 3.5; *p* > 0.05; Figure 3B). Moreover, experimental approaches inhibiting the “Immune Response” resulted in a later onset (N = 19; Chi-square = 10.53; *p* < 0.01; Figure 3C) and extended survival (N = 18; Chi-square = 9.0; *p* < 0.05; Figure 3H), whereas no effects were observed in onset (N = 12; Chi-square = 0.5; *p* > 0.05; Figure 3D) and survival (N = 11; Chi-square = 0.2; *p* > 0.05; Figure 3I) in primary studies employing experimental inhibition of mechanisms linked to “Cell Fate”. Conversely, treatments producing “Trophic Support” have produced delayed symptom onset (N = 27; Chi-square = 32.9; *p* < 0.001; Figure 3E) and extended survival (N = 25; Chi-square = 29.1; *p* < 0.001; Figure 3J).

Then, the data analysis would verify if the described experimental procedures may modify the disease onset and survival time in the same or opposite manner, or if they only affect one of these parameters (Figure 4). The results have shown that experimental procedures inhibiting “Stress” or “Proteinopathy” or providing “Trophic Support” have produced similar effects on disease onset and survival, i.e., by anticipating or delaying both parameters together, in the same primary studies. The Chi-square statistical test confirmed the increase in “Similar” effects and a decrease in both “Opposite” effects and “Only one” parameter, as compared to expected values, for “Stress” (N = 23; Chi-square = 29.6; *p* < 0.001; Figure 4A), “Proteinopathy” (N = 16; Chi-square = 17.0; *p* < 0.001; Figure 4B), and “Trophic Support” (N = 25; Chi-square = 29.9; *p* < 0.001; Figure 4E).

Conversely, anti-inflammatory treatments or other experimental approaches aimed at inhibiting “Immune Response” produced apparently unrelated effects on disease onset and survival (N = 18; Chi-square = 3.7; *p* > 0.05; Figure 4C), and the same was observed for treatments inhibiting “Cell Fate Control” mechanisms (N = 12; Chi-square = 1.8; *p* > 0.05; Figure 4D). When the effects of all experimental procedures were analysed together, they seem to produce similar effects on both onset and survival (N = 94; Chi-square = 68.1; *p* < 0.001; Figure 4F).

### 2.5. Effects of Experimental Procedures on the Pathological Mechanisms

After the analysis of data collected from the primary studies, it was evident that although every single experimental approach should hypothetically act on a specific target (e.g., anti-inflammatory drugs or anti-SOD1 antibodies), a larger series of mechanisms appeared to be modified, because many pathophysiological processes are obviously connected to each other, and the elucidation of the nature of this connection is the main purpose of research efforts.

Unfortunately, the number of primary studies analysed here was not high enough to get reliable information about these complex connections, and further experimental efforts are necessary. However, as the pathological mechanisms most commonly observed in primary studies were those included in “Immune Response” and “Neuromuscular Degeneration” categories, this part of the analysis has focused on the correlation between these mechanisms and all kind of treatments taken together. Figure 5 shows the distribution of primary studies where a treatment (inhibition or stimulation) produced a similar (inhibition or stimulation) or opposite (stimulation or inhibition) effect on the controlled mechanism, if any.

In particular, it appears that a significant correlation exists between treatments and effects on “Immune Response” in more than half of the primary studies (N = 48; Chi-square = 4.0; *p* < 0.05; Figure 5A), whereas the effects of treatments on “Neuromuscular Degeneration” were observed in a high number of studies, but they were apparently highly variable, thus resulting in the lack of correlation (N = 74; Chi-square = 1.8; *p* > 0.05; Figure 5B).

## 3. Discussion

The development of theoretical models explaining ALS etiopathogenesis is important, in order to drive experimental approaches in more promising directions. A number of theoretical studies have so far developed mathematical models describing particular aspects of pathogenesis, such as protein misfolding, aggregation and spreading, metabolism, or oxidative stress [26,27,28], but fewer studies have tried to correlate these aspects with the timing of disease onset and progression, with the aim to develop predictive models capable of identifying key aspects of etiopathogenesis that can be targeted to successfully delay disease onset and progression [18,19,29]. The present meta-analysis study, aimed to demonstrate that a multi-step model can describe the etiopathological development of disease in mouse models of ALS, similar to that previously observed in patient cohorts [18,19]. The results confirmed the hypothesis; thus, despite the obvious differences between patients and animal models overexpressing just one of the at-risk genetic mutations, this evidence supports the emerging paradigm of ALS as a multi-step disease and provides additional predictability to one of the most used ALS mouse models, which is the SOD1 transgenic mouse. It is known that several other genetic mutations are linked to ALS, and some of them are more representative of human disease. However, the number of studies involving these animal models, including those carrying the transactive response DNA binding protein of 43 kDa (TDP-43), fused in sarcoma (FUS) and C9orf72, are currently too limited to allow a reliable quantitative population study.

Importantly, this multi-step paradigm is consistent with the known hypothesis that ALS arises focally by the accumulation and aggregation of misfolding proteins in a small region of the CNS, and such an event could represent one of the steps leading to onset, followed by the failure of autophagic mechanisms, which can represent another step [27,30,31,32,33,34,35,36]. Separate analyses of transgenic mouse models carrying different SOD1 mutations, showed that a different number of steps would be necessary to develop the disease. This finding is in line with a large experience with SOD1 mouse models, showing that an earlier or later onset could be determined by a specific mutation, as well as by copy number [23,24,25]. Sex and genetic background may also affect onset and progression of disease [37,38,39,40], but the large mouse population included in this meta-analysis is balanced for both factors, being a heterogeneous mouse population, similar to patient cohorts. In more detail, after plotting the log incidence against the log age, it was evident that the estimated slope of regression line in mice harbouring the SOD1^G93A^ mutation was of 1, consistent with a two-step process. Thus, if a six-step process would also apply to the animal models, as hypothesized in the present study, then it would be assumed that up to four disease steps are caused by the gene mutation. This finding is in line with the observations of Chiò and colleagues in cohorts of ALS patients with SOD1 mutation [19]. A similar result was observed in the group of mice overexpressing the SOD1^G37R^ mutation, which show an estimated slope of 2, indicating a three-step process. The analysis of mice carrying the SOD1^H46R^ mutation confirmed a multi-step pathological process requiring four steps before manifesting symptoms. Unexpectedly, the mathematical model suggested a six-step disease in the group of mice carrying the SOD1^G85R^ mutation. This finding is surprising, because it is consistent with the previous findings by Al-Chalabi and colleagues [18], showing an estimated six-step process in a large cohort of human patients, regardless of specific ALS-linked mutations (i.e., mainly sporadic ALS patients). Therefore, this finding suggests that the SOD1^G85R^ mutation probably has a low effect on the phenotype, providing a risk that is similar to that observed in an ALS population with no particular genetic predisposition. Interestingly, the average age at onset is apparently correlated with the number of steps towards disease. For instance, SOD1^G93A^ mice showed only two steps, associated to the shorter age at onset (16 weeks), compared to the SOD1^G85R^ mice that showed an average age at onset of about 46 weeks, with an estimated six-step disease process. Therefore, an explanation of these data is that an inverse proportional relationship may exist between the number of steps determined by each mutation and the effect that the genetic lesion exerts on the animal model. The results support the idea that the greater the influence of the gene on the animal model in determining the disease onset, the fewer will be the number of steps in the proposed multi-step model of ALS etiopathogenesis. Furthermore, when comparing the four groups of animals overexpressing the different mutations, it was clear that the higher the average age at onset, the lower the average incidence.

In our view, these results can be explained as follows: although all the animals will surely show disease symptoms because of the overexpression of ALS-linked genes, each mouse model will likely develop the pathological phenotype with a different strength and speed. Noteworthy, we must consider that the different mouse models analysed here have variable levels of mutant SOD1 overexpression in CNS: 17-fold on average for G93A mutation, 2–14-fold for G37R, high expression for H46R and about 1-fold for G85R [23,24,25]. Future studies should extend the analysis to other relevant ALS-linked mutations, but the number of these studies is currently not high enough to allow a meta-analysis study with a sufficient strength. However, the results would suggest that genetic mutations predispose to a greater, but variable, risk of developing the disease, and prove that a specific pathogenic ALS gene could contribute to more than one step, partially explaining the variability of the ALS phenotype. Accordingly, many studies have proposed that the pathogenic process could be already present in humans many years before the appearance of symptoms, but the toxicity of aberrant proteins is a time-dependent cumulative process [6,41,42].

Finally, the results demonstrated a statistically significant correlation between the log of the actual incidence and that predicted by the model, thus confirming the predictive value of this mathematical model. Confirming that a multi-step etiopathological process also occurs in animal models of ALS, as is already known in patients [18,19], could help to identify and characterize every single component of this step-by-step process that triggers the disease. Importantly, future studies should also focus on the identification of non-genetic steps, that might shed light on the etiopathogenesis of ALS in human patients with important impacts on ALS knowledge and therapeutic benefits. Moreover, this finding further supports the increasing evidence that cancer and neurodegenerative disease share epidemiological and pathogenic similarities [17,18,43].

With the aim of contributing to the dissection of pathological determinants of ALS onset and progression, and to the identification of the above-mentioned disease steps, this meta-analysis study also provides a deeper analysis of primary studies, by investigating the relationship between modifications of onset (and animal lifespan) and the underlying cellular and molecular mechanisms. A large number of papers have so far investigated the pathological mechanisms of ALS [3,8] but, to the best of our knowledge, no study to date has addressed the association between these mechanisms and the timing of disease onset and progression in a large population of SOD1^G93A^ mice. The analysis of more than a hundred primary studies has shown here that the experimental inhibition of mechanisms including cellular stress (e.g., those related to apoptosis, glutamate excitotoxicity or oxidative stress) and immune response (e.g., reactive gliosis, inflammatory mediators or white blood cell responses), as well as the activation of trophic mechanisms (including trophic factors or the beneficial trophic effects of cell grafts) may result in both the delay of symptom onset and extension of animal lifespan. Moreover, in order to better understand if the cited mechanisms exert the same actions on both disease onset and progression, their association was studied by correlating them in the same primary studies, thus confirming that stress mechanisms and trophic support does exert similar effects on both parameters, whereas immune mechanisms seem to produce independent effects on symptom onset and survival, despite the cumulative positive action on both aspects of pathogenesis. On the other hand, the inhibition of pathological mechanisms linked to proteinopathy apparently produced similar effects on both parameters, but the cumulative effects on the symptom onset were not significant, whereas a significant extension of survival was found. A series of considerations may arise from these findings. Immune responses include both beneficial and detrimental effects in neurodegenerative diseases such as those dependent on the phenotype of activated microglia [44,45,46]. Therefore, although the inhibition of inflammatory and other immune mechanisms could collectively have average beneficial effects, it is likely that each particular immune process may affect the specific phases of disease in a different manner. The same could apply to the inhibition of proteinopathy-related mechanisms but, apparently, not to the other investigated processes (i.e., cellular stress or trophic mechanisms). It may be argued that the different disease steps can depend on different pathological events, and that the processes leading to the onset of symptoms can be different from those affecting the disease progression. For instance, proteinopathy can supposedly be more important during the later phases of pathological sequence. Similarly, it is possible that certain inflammatory and/or immune processes can be more relevant in the earlier and some other in the later stages, and they could be beneficial in certain disease steps and detrimental in others. Future experimental and theoretical studies are necessary to understand how the described mechanisms may be sequentially linked in a step-by-step progression of pathogenesis towards symptom onset and subsequent disease progression. In particular, the results would suggest that particular attention should be devoted to inflammatory/immune mechanisms and proteinopathy, although the role of the other mechanisms appears to be clearer.

A complex pattern of pathophysiological mechanisms can also explain the lack of a clear effect of treatments acting on a number of molecular pathways linked to the regulation of cell fate, including Notch signalling, antimitotic and antiapoptotic treatments, IRE1α-ASK1 pathway, ephrins, as well as GSK3β, NRG1-Erb, PTEN/Akt, HGF/MET or PDGF signalling pathways. In particular, the inhibition of these pathways could exert a plethora of different functions, probably with opposite final effects on the disease onset and progression, thus confirming the complexity of the molecular network governing the pathophysiology of motor systems [47]. Further studies are necessary to dissect the molecular network underlying the pathological sequence driving ALS, and it could be argued that a system biology approach would be better than the more commonly used reductionist experimental design [5].

It is likely that, among non-genetic steps, an important role could be exerted by plastic changes that continuously compensate for the progressive loss of function during neurodegeneration, thus delaying symptom onset. In fact, symptoms would probably appear when these adaptive changes become unable to overcome the loss of a large number of neurons [10,11,12,13,14,15,16]. So, it could be hypothesized that a sudden failure of compensatory processes may represent one of the unknown steps towards disease. Plastic phenomena can likely be present also after the onset and, though unable to hide motor deficits, they can contribute to increased lifespan. In the animal models analysed here, the lack of correlation observed between treatments and their effects on neurodegenerative processes (motoneuron death, muscle denervation and atrophy) can support the hypothesis of functional adaptations trying to normalize motor activity, despite the loss of large populations of neurons. Existing evidence supports the validity of this approach. For instance, the role of some neurotrophic factors, or the involvement of TDP-43, in synaptic plasticity, together with their known role in cell survival and fate or in RNA processing and pathological involvement, respectively, suggests a complex interaction of these factors [14,48,49,50,51]. Therefore, future studies aimed at elucidating the possible linkage between neuroplasticity and each of the mechanisms underlying ALS etiopathogenesis observed here (inflammation, protein misfolding, cellular stress etc.) would be a promising research direction.

## 4. Materials and Methods

### 4.1. Study Selection and Eligibility Criteria

This study is based on a systematic search of the scientific literature followed by a meta-analysis, with the purpose of calculating a parameter that can be comparable to incidence values from the selected articles employing ALS animal models. SOD1 transgenic mice were chosen, given the relatively higher number of primary studies employing these animals, compared to other more recently discovered mutations. A subset of these primary studies was also used for collecting data about cellular and molecular mechanisms underlying the disease manifestation. The literature search was conducted on PubMed database using the individual search terms “G93A, G37R, G85R and H46R in combination with “amyotrophic lateral sclerosis”, or “mouse disease progression”, or “mouse onset”. Articles published during the last twenty years have been considered and then screened by applying the inclusion criteria. In particular, together with a group of transgenic mice subjected to a disease-modifying treatment or experimental manipulation, the articles had to include a control group of SOD1 transgenic mice without any additional treatment, and a Kaplan–Meier plot showing the distribution of symptom onset in the sample, from which it was possible to extrapolate the incidence data.

### 4.2. Calculation of Incidence Data

The first phase of data collection was the analysis of Kaplan–Meier curves to obtain the number of animals showing symptom onset at a defined age, including in the analysis of only untreated SOD1 mice. After applying the same procedure to each study, monthly incidence values were obtained by dividing the number of mice showing symptoms during a defined month by the total number of mice being still asymptomatic (susceptible) in the same month. Considering the differences in life expectancy and in disease duration between mice and human patients and considering the number of cases per time-point, it appeared that a monthly estimation could be a good predictor of incidence in mouse models. Once the incidence was calculated as described, the mathematical model described below was applied to each mutation considered in the study.

### 4.3. Multi-Step Model

The mathematical model used to determine if ALS is a multi-step process was originally described by Armitage and Doll [17] in their multi-stage cancer theory, and recently adapted to ALS by Al-Chalabi and colleagues (2014). According to this multi-step hypothesis, if we suppose that ALS is caused in one step, then the incidence (i) of ALS in human patients will be proportional to the risk (u) of undergoing the defined step in that year; therefore, the incidence depends on both risk and time. Otherwise, if the disease takes more than one step (each step with risk u_i_) then the probability of being exposed to the first step at t years is u_1_t; the second step by age t years is u_2_t as long as the final disease-causing step is achieved (the n − 1 step). This process can be explained by the following formula in which incidence, in a given year, is related to the risk (u) [17,18]:i = u_1_u_2_ … u_n−1_ u_n_t^n−1^

Plotting the log of incidence against the log of age will result in a linear relationship:log(i) = (n − 1)log(t + c)
where n, the slope of the regression line, is one less than the number of steps needed to develop ALS, while the intercept (c) describes the probability of undergoing these steps.

Here, the same mathematical model has been applied by using the monthly incidence calculated as described above. This way, if a multi-step disease concept does occur in animal models, then a linear regression between the log incidence and the log age at onset would occur, and the slope of the regression line can be used to estimate the number of steps.

### 4.4. Extraction of Data concerning Pathological Mechanisms and Effects of Treatments

The subset of primary studies employing SOD1^G93A^ mice were also used to collect data concerning the cellular and molecular mechanisms associated with the modifications of onset and survival time shown in each primary study. Only mechanisms that were described as significantly modified were considered and classified as increased or decreased. Similarly, data concerning the treatments or experimental manipulations used in primary studies, which were able to affect onset and/or survival, and the described underlying mechanisms, were also collected and classified as stimulatory or inhibiting treatments. Also, data concerning the modifications of onset and survival were collected and classified as earlier or later (or unchanged) onset and reduced or extended (or unchanged) survival. These data were then analysed to search for any association between onset (or survival) and their determinants, or for associations between treatments and their putative effects on onset (or survival) and their cellular and molecular determinants.

### 4.5. Statistics

Data were collected and analysed using Microsoft Excel. Statistical analysis was done by using Systat. The validity of the multi-step model was determined by linear regression and correlation of logarithmic values of both incidence and age at onset. The association between treatments and their effects on onset and survival, or between onset (or survival) and their underlying mechanisms were estimated by using the Chi-square test. Null hypothesis was discarded with *p*-values < 0.05 or less.

## 5. Conclusions

The present meta-analysis study has analysed data extracted from 137 primary studies, involving over 4600 mice carrying a mutation of the SOD1 gene. The purpose of the study was twofold: the first aim was to set up a population-based modelling study of the onset of symptoms in a large population of mice, in order to verify if the etiopathological mechanism leading to the disease is a multi-step process, as recently shown by epidemiological studies involving large cohorts of ALS patients. The results have confirmed the hypothesis, showing that the appearance of symptoms in SOD1 mutant animal models requires a series of two to six sequential steps, depending on the particular gene mutation. These findings are in accordance with those previously shown in human populations carrying an ALS-linked gene mutation. The second aim was to evaluate the association between the etiopathological mechanisms of disease and their contribution to the timing of onset and progression of motor symptoms, in order to provide a contribution to the identification of the above-mentioned steps leading to the disease. The results have shown that a large plethora of pathological processes identified by primary studies may influence both age at onset and survival (i.e., cellular stress and trophic mechanisms), although proteinopathy apparently has stronger effects on survival time. On the other hand, inflammatory and immune mechanisms may exert complex effects, and it is possible that some can be more relevant in the earlier and some other in the later stages, and they could be beneficial in certain disease steps and detrimental in others. Finally, data support the idea that compensatory plastic changes may be part of the described multi-stage pathological process. Collectively, the data shown here provide, in our view, further evidence about the predictability of ALS animal models and their usefulness as tools to dissect the etiopathogenesis of this disease and to validate novel therapy approaches and targets. Future extension of the present study would include other relevant animal models of ALS, in order to include more etiopathological aspects in the mathematical model.

## Figures and Tables

**Figure 1 ijms-24-00216-f001:**
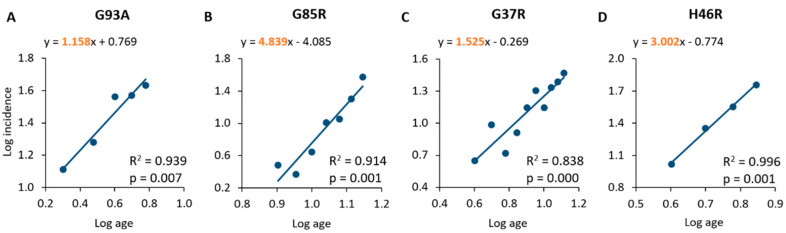
Linear regression and correlation of log age and log incidence in mouse populations of SOD1^G93A^ (**A**), SOD1^G85R^ (**B**), SOD1^G37R^ (**C**) and SOD1^H46R^ (**D**). The equations of the lines representing the best fit of linear regression are shown in the graphs. The slopes are highlighted in orange.

**Figure 2 ijms-24-00216-f002:**
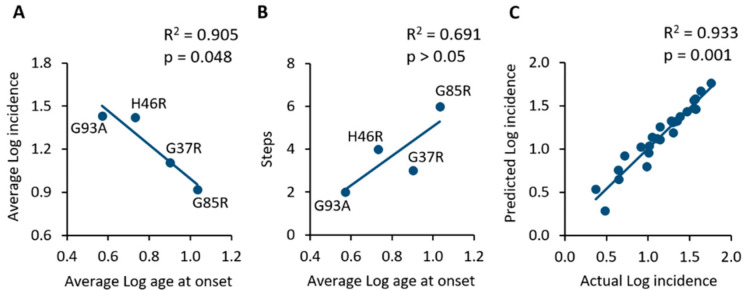
(**A**) Comparison of SOD1 mutant mouse populations showing an inverse correlation between the average age at onset and the average incidence in the same population. (**B**) The application of the Armitage and Doll model suggests that the lower the number of steps towards onset, the earlier the average age at onset of animals. (**C**) Linear regression and correlation of the actual incidence values extracted from primary studies and those predicted by the mathematical model. The strong correlation suggests the high predictability of the model.

**Figure 3 ijms-24-00216-f003:**
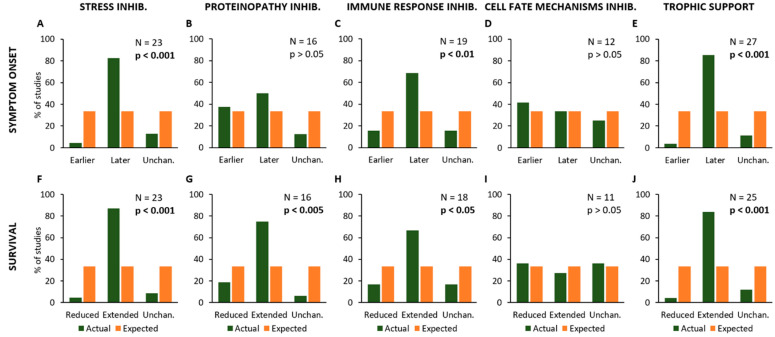
Frequency distribution of the effects of different treatments or experimental manipulations on the onset of symptoms (**A**–**E**) and survival time (**F**–**J**). The graphs show the percent number of studies where a given kind of treatment or manipulation is associated with a modification of onset and/or animal lifespan (in green), compared to the expected values (null hypothesis, in orange). In particular, the modifications of onset (anticipation or delay) and survival time (reduction or extension) are shown in relation to the treatments aiming at reducing Stress (**A**,**F**), Proteinopathy (**B**,**G**), Immune Response (**C**,**H**) and Cell Fate control (**D**,**I**), or providing Trophic Support (**E**,**J**). Each graph shows the total number of studies where a given kind of treatment/manipulation has been evaluated and the *p*-value calculated by using the Chi-square statistical test, representing the probability of null hypothesis when comparing the actual values to the expected ones.

**Figure 4 ijms-24-00216-f004:**
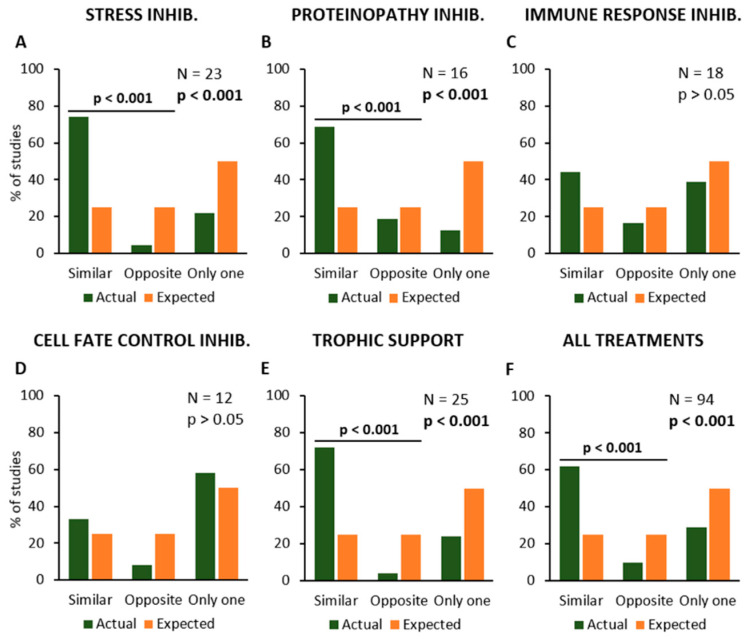
Frequency distribution of the primary studies showing similar or opposite modification of symptom onset and survival time, or the modification of only one parameter, in response to the different types of treatment or experimental manipulation. Percent number of studies where the effect was observed (in green), are compared to the expected values (null hypothesis, in orange). The modifications of onset (anticipation or delay) and survival time (reduction or extension) are shown in relation to the inhibition of Stress (**A**), Proteinopathy (**B**), Immune Response (**C**) and Cell Fate Control mechanisms (**D**), and stimulation of Trophic Support (**E**). The average effects of all treatments together are shown in (**F**). Each graph shows the total number of studies where the effects of a given kind of treatment or manipulation has been evaluated and the *p*-value calculated by using the Chi-square statistical test. The horizontal bars and the relative *p*-values are relative to a Chi-square test comparing similar and opposite effects, with the exclusion of studies reporting effects on only one parameter (onset or survival).

**Figure 5 ijms-24-00216-f005:**
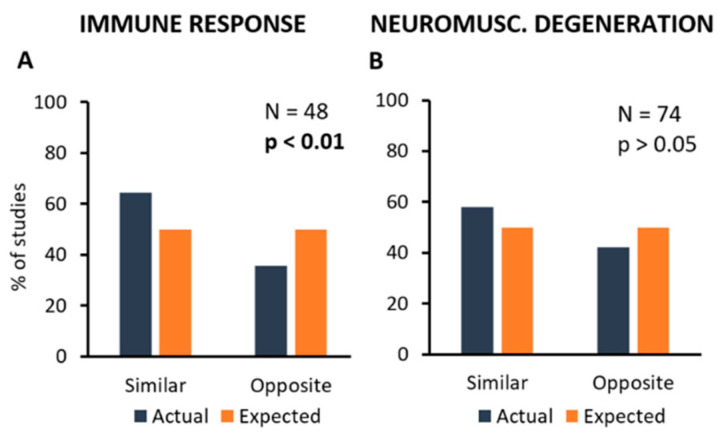
Frequency distribution of the primary studies showing the association between treatments and the mechanisms they can modify. In particular, the most frequent types of mechanism, i.e., Immune Response (**A**) and Neuromuscular Degeneration (**B**) have been considered. Percent number of experiments showing the modification of one of these mechanisms (in blue) are compared to the expected values (null hypothesis, in orange). Each graph shows the total number of experiments analysing a particular mechanism, and the *p*-value calculated by using the Chi-square statistical test.

**Table 1 ijms-24-00216-t001:** Number of primary studies found by PubMed search, and selected for meta-analysis after application of inclusion criteria, with the corresponding number of mice used for incidence analysis.

SOD1 Mutation	Total No. of Papers	No. of Excluded Papers	No. of Selected Papers	No. of Mice
G93A	639	530	109	2073
G85R	29	18	11	479
G37R	38	26	12	314
H46R	19	14	5	172
Totals	725	588	137	3038

**Table 2 ijms-24-00216-t002:** Incidence data calculated for each SOD1 mutation, for the application of the Armitage and Doll mathematical model.

SOD1 Mutation	Month	No. of Cases	No. of Susceptible Subjects	Monthly Incidence (%)	Log Age	Log Incidence
G93A	2	35	272	12.9	0.30	1.11
	3	476	2498	19.1	0.48	1.28
	4	1018	2798	36.4	0.60	1.56
	5	380	1025	37.1	0.70	1.57
	6	78	183	42.9	0.78	1.63
G85R	8	6	198	3.0	0.90	0.48
	9	31	1326	2.3	0.95	0.37
	10	62	1404	4.4	1.00	0.65
	11	129	1258	10.3	1.04	1.01
	12	106	943	11.2	1.08	1.05
	13	86	430	20.0	1.11	1.30
	14	47	125	37.6	1.15	1.58
G37R	4	21	469	4.5	0.60	0.65
	5	39	402	9.7	0.70	0.99
	6	19	362	5.2	0.78	0.72
	7	29	355	8.2	0.85	0.91
	8	56	401	14.0	0.90	1.15
	9	37	183	20.2	0.95	1.31
	10	16	114	14.0	1.00	1.15
	11	24	111	21.6	1.04	1.33
	12	32	131	24.4	1.08	1.39
	13	13	44	29.5	1.11	1.47
H46R	4	10	96	10.4	0.60	1.02
	5	65	288	22.6	0.70	1.35
	6	44	123	35.8	0.78	1.55
	7	8	14	57.6	0.85	1.76

**Table 3 ijms-24-00216-t003:** Comparison of the mouse populations overexpressing the SOD1 mutations.

SOD1 Mutation	Average Age at Onset (Weeks)	Log Average Age at Onset	Average Incidence (%)	Log Average Incidence	Slope	Steps
G93A	16.0	0.57	29.6	1.43	1.16	2
G85R	46.4	1.03	12.7	0.92	4.84	6
G37R	32.8	0.90	15.1	1.11	1.53	3
H46R	21.6	0.73	31.5	1.42	3.00	4

**Table 4 ijms-24-00216-t004:** Classifications of pathological mechanisms and experimental procedures in categories.

Experimental Procedures	Main Pathological Mechanisms
Stress inhibition	Immune Response
Proteinopathy inhibition Immune Response inhibition	Neuromuscular Degeneration
Cell Fate Mechanisms inhibition	
Trophic Support	

## Data Availability

The data presented in this study are available on request from the corresponding author.

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
