# Peer review of "A Meta-Analysis Study of SOD1-Mutant Mouse Models of ALS to Analyse the Determinants of Disease Onset and Progression"

_ijms, 2022, doi:10.3390/ijms24010216_

Round 1

Reviewer 1 Report

The paper by Ciuro et al. report a meta-analysis of preclinical studies done in SOD1 mutant mouse models to explore the determinant of disease onset and progression.

The study is particularly interesting, considering the large amount of data available for the mutant SOD1 mouse model and very limited systematic information available at the moment. However, there are some aspects that have to be clarified and data that are missing. Without this information it is difficult to judge the reliability and relevance of the analysis. It is important to state that the onset and progression of the disease depends not only from the type of SOD1 mutation, but also from the genetic background, copy number of the transgene and sex. Several studies have been published on this issue, especially on the G93A SOD1 mutant that show huge variability in the onset and progression of the disease because of these parameters, for example Pfohl et al., 2015 and Marino et al., 2015. For these reasons a more detailed characterization of the mice and of the studies included in the analysis should be provided in the paper.

These are the major points that have to be considered:

1) The papers included in the analysis should be reported as references.

2) More information on the mice included in the study should be given:

·        - % of the studies without sex information

·        -Genetic background of the mice SOD1G93A, for example % of studies using the most common strains: BL6SJL and C57BL/6

·         -Report some information on level of expression of the transgene, only high expressor or also low expressor? Which %?

·         -What are the characteristics of the mice excluded from the analysis (those with very ealier or later onset? Is it for the genetic background or transgene expression level?  

·         -Specify which papers are included in the different categories of “experimental procedures” and “main pathological mechanisms”

Author Response

The paper by Ciuro et al. report a meta-analysis of preclinical studies done in SOD1 mutant mouse models to explore the determinant of disease onset and progression.

The study is particularly interesting, considering the large amount of data available for the mutant SOD1 mouse model and very limited systematic information available at the moment. However, there are some aspects that have to be clarified and data that are missing. Without this information it is difficult to judge the reliability and relevance of the analysis. It is important to state that the onset and progression of the disease depends not only from the type of SOD1 mutation, but also from the genetic background, copy number of the transgene and sex.

Response: the authors wish to thank the Reviewer for the careful evaluation of the manuscript and for considering the originality of this work. As requested by the Reviewer, additional data have been included in the revised version of the manuscript, and the details are described below.

Several studies have been published on this issue, especially on the G93A SOD1 mutant that show huge variability in the onset and progression of the disease because of these parameters, for example Pfohl et al., 2015 and Marino et al., 2015. For these reasons a more detailed characterization of the mice and of the studies included in the analysis should be provided in the paper.

Response: as also explained below, more precise indications concerning the impact of transgene expression, sex and strain of SOD1 mice have now been included and discussed in the manuscript. Other details about the primary studies included in the analysis are also added in the revised version. The authors also wish to thank the Reviewer for suggesting the papers by Pfohl et al., and Marino et al., which are helpful for addressing these concerns and improving the quality of the study.

These are the major points that have to be considered:

1) The papers included in the analysis should be reported as references.

Response: a list of the papers selected for this metanalysis has been included in the Supplementary Materials section. Since these papers were not cited in the main text, they have not been included in the main text References.

2) More information on the mice included in the study should be given:

  •       - % of the studies without sex information

Response: the % of studies involving male, or female mice, or using a sex balanced protocol, or otherwise not declaring information about the sex of mice in the experimental design have now been included in the Results, section 2.1, and briefly commented in Discussion. In particular, only 28% of studies showed separate analyses of female and/or male mice; while 33% used a sex balanced protocol and 39% of studies did not state this information.

  • -Genetic background of the mice SOD1G93A, for example % of studies using the most common strains: BL6SJL and C57BL/6

Response: the genetic background of SOD1 transgenic mice appeared to be balanced across primary studies: 38% and 45% for B6SJL and C57BL/6, respectively, whereas 17% of studies did not include strain information. These data have been included in the section 2.1 of Results and briefly commented in Discussion.

  •        -Report some information on level of expression of the transgene, only high expressor or also low expressor? Which %?

Response: this comment is important and it gives us the opportunity to extend the discussion about the role of copy number of the SOD1 transgene in animal models. The role of this factor was already discussed (please see lines 358-360 and 391-393 of Discussion in the first version of the manuscript), but it might be further clarified. A deeper analysis of statements found in the primary studies has confirmed that SOD1-G37R mice had a 2-14-fold expression of the transgene, whereas SOD1-G85R mice are low-expressors (although 50% of papers employing this model did not declare the copy number or SOD1 protein level). Conversely, SOD1-H46R are high-expressors. Concerning SOD1-G93A mice, we found that 87% of primary studies employed high-expressors, while 5% used low-expressors and 8% did not declare the mouse strain. These data are now reported in the revised version of the manuscript (please see Results, section 2.1, and Discussion).

  •        -What are the characteristics of the mice excluded from the analysis (those with very ealier or later onset? Is it for the genetic background or transgene expression level?  

Response:  only 10, out of 137 papers included a small number of mice that was excluded from the study due to a very earlier or later onset (accounting for less then 5% of the entire population, as stated in the first version of the manuscript). Among these studies, only 1 included mice with a very early onset, and those mice were high SOD1 expressors, with a C57BL/6 genetic background. The remaining 9 papers included some mice showing very later onset. Among the latter, 5 studies included high-expressors; 3 included low-expressors and 1 did not state this information. These mice showing a very late onset had a C57BL/6 genetic background in 2 cases and B6SJL in 3 cases (4 papers did not state the genetic background). From these data, it seems that neither the genetic background or the transgene expression level significantly influenced the behaviour of these outliers. A brief statement concerning these findings is now reported in the Results, section 2.2.

  •        -Specify which papers are included in the different categories of “experimental procedures” and “main pathological mechanisms”

Response: a table containing the reference numbers of the papers included in these categories is now included in Supplementary materials (Supplementary table 1) and cited in Results, section 2.3.

Reviewer 2 Report

Cuiro et al. performed a metaanalysis of available data for various mutant SOD1 ALS mouse models to determine onset and survival as well as pathological mechansims using a mathematical model. Although interesting as an approach several major issues remain unclear throughout the study.

1. Why only mutations of SOD1 were selected and their mouse models? What about other effected genes and the data gathered with their mouse models? Surely other models with other genes affected could have provided an more complete picture of the pathology, independent if they may be in lower numbers compared to SOD1 mouse models. It is known that other mouse models might not primarily affect motoneurons but e.g. oligodendrocytes (example OPTN) which then leads to death of motoneurons through a non-cell autonomous mechanim, In this context the the study presented seems very reducionistic and its not clear what is the final aim, since it appears not aimed to explain disease mechanisms. Since the current picture appears a lot more complex than just focusing on SOD1. The notion of the authors in the Abstract that SOD1 models appear to model well the disease appears automatic since SOD1 is the oldest model and the majority of data available comes from the model, yet it cannot automaticly be assumed that its the best representation of the disease and the potential mechanism identified with it. 

2. Beside the above, it remains elusive if the genetic background of the SOD1 mouse models was considered in the selection of the studies, since it is well known that depending of the background of the SOD1 animals differences of onset and survival will be found (e.g. C57BL6 vs B6SJL). 

3. It is not clear what is the main contribution and conclusion of the work since it states that even with the SOD1 model it does not have enough numbers of studies. Also, it is not clear how the information provided is useful to understand the disease. 

4. It would have been interesting to see a compilation of studies done with other genes affected and compare them with SOD1 models and in this way assess the generality of SOD1 as a representative for the disease. 

Author Response

Cuiro et al. performed a metaanalysis of available data for various mutant SOD1 ALS mouse models to determine onset and survival as well as pathological mechansims using a mathematical model. Although interesting as an approach several major issues remain unclear throughout the study.

Response: the authors would like to thank the Reviewer for her/his interest in the present study. A point-by-point reply to the Reviewers’ comments is reported below.

  1. Why only mutations of SOD1 were selected and their mouse models? What about other effected genes and the data gathered with their mouse models? Surely other models with other genes affected could have provided an more complete picture of the pathology, independent if they may be in lower numbers compared to SOD1 mouse models. It is known that other mouse models might not primarily affect motoneurons but e.g. oligodendrocytes (example OPTN) which then leads to death of motoneurons through a non-cell autonomous mechanim, In this context the the study presented seems very reducionistic and its not clear what is the final aim, since it appears not aimed to explain disease mechanisms. Since the current picture appears a lot more complex than just focusing on SOD1. The notion of the authors in the Abstract that SOD1 models appear to model well the disease appears automatic since SOD1 is the oldest model and the majority of data available comes from the model, yet it cannot automaticly be assumed that its the best representation of the disease and the potential mechanism identified with it.

Response: we would like to thank the Reviewer for this comment, which gives us the opportunity to further clarify the purpose of this metanalysis study. In particular, this work moved from two interesting epidemiological studies (Al-Chalabi et al., 2014, Chiò et al., 2018) showing, for the first time, that ALS is not only a multifactorial disease, where genetic and environmental factors interact in a complex manner, but also a multi-step disease, where symptom onset occurs only after patients have been exposed to a sequential series of a precise number (six) of risk factors. Among these factors (to be identified), genetic mutations were found to account for certain (from two to four) but not all of these steps. Identifying these (six) etiopathological, sequential steps, remains an important challenge. Thus, since animal models are largely used in mechanistic and therapeutic experimental protocols, we proposed to verify if this multi-step paradigm already found in patient populations could also apply to animal models. To this aim, large population of mice (thousands) were required to make a population study and to estimate disease incidence. This is the main reason for choosing the SOD1 transgenic mice (as already stated in Methods, section 4.1), the second one being that this model has been the mainly used so far for mechanistic and therapeutic studies. It is however important to note, as this Reviewer does, that a number of other genes were found to be related to ALS, and that some of them are more representative than SOD1 models, including animal models of TDP-43, FUS or C9orf72, and that a large number of other genes are also involved in ALS pathogenesis, including for example OPTN and many others. However, this work was not intended as a comprehensive modelling study aiming at mimicking all aspect of ALS, as this purpose would probably be too ambitious and, however, out of the scope of the present study. Moreover, as stated in Methods, the number of studies (and the number of mice included) involving other (although important) mutations are still too small to allow a population study with reliable quantitative results, but future studies would address these aspects. Some additional statements have been included in the revised manuscript to address these important questions; please see Discussion and Conclusions in particular.

  1. Beside the above, it remains elusive if the genetic background of the SOD1 mouse models was considered in the selection of the studies, since it is well known that depending of the background of the SOD1 animals differences of onset and survival will be found (e.g. C57BL6 vs B6SJL).

Response: we thank the reviewer for this comment. The impact of the genetic background is known and a couple of references have now been included to help describing and discussing this aspect. However, the contribution of genetic background was not included in the mathematical model, because it was out of the scopes of this study and it would reduce the number of mice in each group, thus reducing the power of this population study. On the other hand, as also stated above, one of the main purposes of the present study was to analyse a large population of mice (over 3000 mice were included) in order to verify if the incidence of ALS-like symptoms may follow a multi-step model, as already found in patients’ cohorts (Al-Chalabi et al., 2014, Chiò et al., 2018). With this aim, in our view, it would be important to include different disease-modifying variables into the population, including sex and genetic background, since patient populations were also heterogeneous concerning these variables. However, in this revised version of the manuscript, we include a characterization of the mouse population analysed in the study, thus showing that the genetic background of SOD1 transgenic mice appeared to be balanced across primary studies: 38% and 45% for B6SJL and C57BL/6, respectively, whereas 17% of studies did not include strain information. These data, together with information about sex, have been included in the section 2.1 of Results and briefly commented in Discussion.

  1. It is not clear what is the main contribution and conclusion of the work since it states that even with the SOD1 model it does not have enough numbers of studies. Also, it is not clear how the information provided is useful to understand the disease.

Response: In Results, section 2.1, we already stated that: “After exclusion of the articles that did not comply with the inclusion criteria, 137 papers were selected, and these studies involved a significantly high total number of mice to produce a reliable analysis of data. Precisely, for the analysis of incidence data, a total number of 3.038 control SOD1 transgenic mice (without any other experimental manipulation) were found included in the selected primary studies, with an average number of 22 mice per paper.” Conversely, when we tried to collect primary experimental studies involving other animal models, from the literature, we found a limited number of studies, involving a limited number of mice, that was not high enough to produce reliable results. As stated in the Conclusions, future studies would hopefully extend the results provided here. Concerning the contribution of the present study to the elucidation of ALS pathogenesis, we stated that this is the first study involving a large number of mice in a population-based analysis, aiming at verifying if the etiopathogenesis of ALS could occur in a multi-step way, as recently shown in patient populations, and trying to contribute to the initial identification of these steps. This is an important finding, although limited to the SOD1 animal model, because these data would support the validity of this animal model and can serve for future studies aiming at dissecting the molecular nature of the multi-step pathogenic process. We hope that the purposes of the study and the conclusions are cleared in the revised version.

  1. It would have been interesting to see a compilation of studies done with other genes affected and compare them with SOD1 models and in this way assess the generality of SOD1 as a representative for the disease.

Response: this is interesting, but at the moment this is out of the scope of the present study. Please see also the response to the question no. 1 and 3 for details.

Reviewer 3 Report

“A metanalysis study of SOD1-mutant mouse models of ALS to 2 analyse the determinants of disease onset and progression” is an interesting study for ALS research. This manuscript can be accepted in IJMS after major revision.  

Here are the points:

“multi-step disease” is not a correct word. Instead, authors can use “multifactorial”. Because ALS is not a step by step disease.

Abstract could include more quantitative results of the study.

It is not clear how they chose the 725 papers and then decided 137 papers. 

In Figues 3,4 and 5, the significancy is not clear in the graphs.

Authors should cite “Int. J. Mol. Sci. 2022, 23(5), 2400; https://doi.org/10.3390/ijms23052400” recently published paper in IJMS due to the similarity in particular in Introduction.

Author Response

“A metanalysis study of SOD1-mutant mouse models of ALS to 2 analyse the determinants of disease onset and progression” is an interesting study for ALS research. This manuscript can be accepted in IJMS after major revision.  

Response: the authors would like to thank the Reviewer for her/his interest in the present study. A point-by-point reply to the Reviewers’ comments is reported below.

Here are the points:

  • “multi-step disease” is not a correct word. Instead, authors can use “multifactorial”. Because ALS is not a step by step disease.

Response: this comment by the Reviewer gives us the opportunity to better clarify the purpose of the study. It is in fact well-known that ALS is a multifactorial disease where multiple genetic and environmental cues cooperate to cause the pathological microenvironment, and this was already stated in the first lines of Introduction. However, the term “multi-step disease” is referred to recent epidemiological data supporting the idea that ALS does appear as a step-by-step disease, like previously found for cancer diseases. In particular, two important papers reporting epidemiological analyses of large cohorts of ALS patients [Al-Chalabi et al., Analysis of amyotrophic lateral sclerosis as a multistep process: a population-based modelling study. Lancet Neurol. 2014; Chiò et al., The multistep hypothesis revisited: the role of genetic mutations. Neurology 2018, respectively ref. no. 16 and 17 in the first version of the manuscript], by using the mathematical model developed by Armitage and Doll for cancer (ref. no. 15 in the first version of the manuscript and no. 17 in the revised version; please see Introduction and Discussion), have found that ALS could be a six-step disease, where symptom onset occurs only after patients are sequentially exposed to six (to be identified) risk factors. Genetic risk factors may account for certain, but not all, of these steps. One of the main purposes of the present metanalysis study was to verify if this multi-step hypothesis could also apply to animal models of ALS, and this idea would be of interest to the authors given that a large number of mechanistic and pharmacological studies involve animal models. The authors tried to better explain these aspects in the Introduction and Discussion.

  • Abstract could include more quantitative results of the study.

Response: given the complexity of quantitative data, only the following numeric results have been included in the abstract, to avoid an increase of abstract length and complexity: i) number of primary studies selected for incidence study and for mechanistic analysis (the main purposes of the metanalysis); ii) the number of steps calculated by applying the mathematical model.

  • It is not clear how they chose the 725 papers and then decided 137 papers. 

Response: the criteria used for the selection of literature and the inclusion criteria adopted for selecting the studies to be included in the metanalysis were described in the Methods and Results. We tried to better clarify these criteria in the revised version of the manuscript, in Methods, section 4.1.

  • In Figues 3,4 and 5, the significancy is not clear in the graphs.

Response: please note that the graphs in figures 3, 4 and 5 do not show average values measured in experimental samples, but the actual frequency distribution of primary studies showing a given effect of a treatment onto the age at onset or animal lifespan (in green or blue) compared to the expected frequency (in case of null hypothesis, in orange). Thus, statistical analysis has been done by using the Chi-square test, and p-values are referred to the probability that actual frequencies in all categories of studies (showing earlier or later onset or death) are similar to the expected ones (null hypothesis). So, the p-values reported in the graphs are referred to the whole set of studies included in each category (Stress, Proteinopathy, and so on) and no other symbols showing significance have to be added. In addition, figure 4 also reports the p-values relative to a Chi-square test evaluating the frequency distribution of studies showing only similar or opposite effects on onset and survival (horizontal bars and relative p-values), excluding those showing independent effects on these parameters (only one). In the revised version of the manuscript, we tried to better explain these statistical aspects in the figure captions.

  • Authors should cite “Int. J. Mol. Sci. 2022, 23(5), 2400; https://doi.org/10.3390/ijms23052400” recently published paper in IJMS due to the similarity in particular in Introduction.

Response: this suggestion gives authors a chance to cite the current status of therapeutic approach, an aspect that was mistakenly omitted in the first version of the manuscript. This one, and another (Feldman et al., 2022) recent review, have been cited in the first lines of Introduction.

Round 2

Reviewer 3 Report

The authors have satisfactorily addressed all of my concerns.